# Effect of Internet Use to Obtain News on Rural Residents’ Satisfaction with the Social Environment: Evidence from China

**DOI:** 10.3390/ijerph20031844

**Published:** 2023-01-19

**Authors:** Yusong Liu, Wenrong Qian, Linyi Zheng

**Affiliations:** China Academy for Rural Development, School of Public Affairs, Zhejiang University, Hangzhou 310058, China

**Keywords:** information acquisition method, internet, social environment satisfaction, digital divide, CFPS

## Abstract

The deterioration of satisfaction with the social environment in rural areas recently has become a major issue in the rural governance of China. This study examined if using the Internet to obtain news would affect rural residents’ satisfaction with their social environment. It used data from the China Family Panel Studies to determine the influencing factors of rural residents’ satisfaction with their social environment in the Internet era. The research found that: (1) The Internet has become the main medium for rural residents to obtain news, which affects rural residents’ satisfaction with the social environment. Specifically, as the number of days rural residents use the Internet to obtain weekly news increased, their satisfaction with China’s social environment worsened. Internet use also affected satisfaction with the social environment more than traditional media, such as TV. (2) The influencing factors of rural residents’ social environment satisfaction were heterogeneous among groups with different characteristics, with the phenomenon of the “digital divide” emerging. Women, middle-aged and older adults, and illiterate/semi-illiterate people used the Internet to obtain news less frequently. Based on the above research conclusions, governments should focus on and be vigilant against false public opinions disseminated online as well as improve the digital literacy of vulnerable rural groups.

## 1. Introduction

The new generation of the information revolution dominated by the Internet has advanced rapidly, not only changing the lifestyles of urban residents but also having a huge impact on the lives of rural residents. Empowering rural governance through digital and intelligent means can effectively promote the development of the local economy, improve the happiness of rural residents, and foster a more balanced and orderly rural lifestyle. Without stepping outside their homes, rural residents can use the Internet to handle various government affairs, buy and sell goods, and promote tourism, which has tremendously increased convenience for many rural residents. As the largest developing country in the world, China’s rural Internet has developed rapidly in recent years. According to the 49th “Statistical Report on China’s Internet Development” issued by the China Internet Network Information Center (CNNIC), the number of Internet users in China had reached 1.032 billion as of December 2021, an increase of 42.96 million compared with December 2020. The Internet penetration rate had reached 73.0%, and the average weekly online time per capita was 28.5 h. There were 284 million Chinese rural netizens, accounting for 27.6% of the total netizens, while the Internet penetration rate in rural areas was 57.6%. Due to its timeliness and convenience, an increasing number of rural residents are turning to the Internet to obtain information and express their demands [1,2,3,4,5,6,7]. However, the popularity of Internet technology can also be a double-edged sword.

Compared with the closed, one-way transmission of traditional media, the Internet, as a new media, is an open and diverse platform. Therefore, news and public opinion disseminated on the Internet could significantly affect rural residents’ satisfaction with the social environment. In recent years, many major online news events in China have grown on the Internet and rapidly expanded their scope of influence, becoming well-known news sources, gaining societal attention, and improving governance [8]. Similarly, if the government’s governance model is to change from a ruling model to a service model, it is imperative to spend time understanding the real needs of rural residents [9,10].

Local rural residents have the most say in the government’s implementation of people’s livelihood policies in rural governance. Through on-the-ground research on the satisfaction of rural residents with the social environment, it is possible to directly test the effectiveness of government work at the grassroots level, as social stability ultimately depends on citizens’ satisfaction with society [11]. Therefore, what is the situation of Chinese rural residents’ satisfaction with the social environment? Does Internet use affect rural residents’ social environment satisfaction, and is this impact positive or negative? These issues have important implications for governance in developing countries.

Currently, academic research on rural residents’ satisfaction with the social environment is relatively sparse, and only a few scholars have researched specific fields. These include Liang’s research on government integrity and fiscal transparency; Jiang’s research on the quality of life of urban residents; Shi and Cui’s research on social justice and life satisfaction; Yang and Zhou’s research on educational equity and educational modernization; Xie and Zhao’s research on the medical security system; Tang and Chen’s research on the ecological environment; Wang’s research on satisfaction with basic housing security services; and Wu’s Research on China’s Social Security [12,13,14,15,16,17,18,19]. Kögel determined that public health effects primarily relate to the environment, public safety, lifestyle, and socio-economic and political contexts. In general, scholars are more aware of the significant impact of the Internet on rural residents’ decision-making [20]. For example, Pan believed that the Internet could effectively overcome their information disadvantages [3]. Do et al. reported a high prevalence of frequent Internet use among Vietnamese young people aged 16 to 30. Among the 1200 participants, almost 65% of them used the Internet every day [21]. Wang and Xie pointed out that the information search behavior of tourism websites has become an important part of research on tourist behavior decision-making, while Li and Zhang asserted that tourists who search for information through the Internet spend more money at the destination than tourists who obtain information through other channels [4,22]. Feng and Yao pointed out that consumers’ online products will affect consumers’ purchasing decisions, and Bhalchandra studied the influence of the Internet on patients’ choice of total knee replacement surgery and surgeon [7,23].

A few studies have examined the Impact of Internet use on satisfaction with the social environment. It has only begun to receive attention in recent years due to the rapid development of the Internet, and only some scholars have adequately explored similar concepts of satisfaction with the social environment, such as residential happiness. For instance, Hall found that those who spent less time online, less time expressing emotions, and more time fact-checking scored higher in terms of well-being [24]. Those with higher happiness scores were also more likely to view online disinformation as offensive. Chae conducted an analysis using a sample book from South Korea to re-examine the relationship between social media use and happiness, with the findings suggesting that information on social media may lead us to believe that, through social comparison, individuals feel that other people are better off, happier than they are, and thus are less happy about themselves [25]. Ma and Le studied the impact on the happiness of rural residents, arguing that the Internet has significantly improved the happiness of rural residents through at least two channels—online learning and leisure and entertainment [26].

By combing the literature, we found that the previous research had a few shortcomings. First, the quantitative analyses were mostly based on regional surveys, with no analysis of national samples. Hence, there may be sample selection bias. Second, most surveys of rural residents’ satisfaction with the social environment focus on certain areas, such as medical care, education, and the gap between the rich and the poor, without considering the overall situation. Third, related studies did not consider the differences in the characteristics of different groups of people. Due to historical and geographical factors, certain differences exist in the economic, social, and cultural development of different rural areas in China. The development gap will inevitably lead to differences in satisfaction with the social environment and information acquisition methods among rural residents with different characteristics. For example, Cui and Feng noted that in most rural areas of China, especially marginal areas and those with old, young, and poor individuals, the problem of insufficient information for farmers has not been completely solved [27]. Accordingly, based on the CFPS2014 national survey data, this study will use econometric methods to analyze the impact of modern media, such as the Internet and TV, on rural residents’ satisfaction with the social environment, as well as differences in information acquisition methods among rural residents with different characteristics, analyze the deep-seated reasons, and propose corresponding countermeasures.

## 2. Research Framework and Hypothesis

According to the rationality principle in economics, in normal circumstances, rural residents are inclined to evaluate the satisfaction of the social environment owing to specific reasons and make the most favorable judgment for themselves under the simultaneous influence of internal and external factors:The influence of internal factors—Referring to the research of scholars, the personal characteristics of rural residents will affect the results of social environment satisfaction, such as gender, age, ethnicity, marital status, income, working status, education level, health status, and other factors [17,28].The influence of external factors—Due to the presence of externalities, individuals will be directly affected by the social environment in which they live, examples include government corruption, inequality between the rich and poor, environmental protection, employment, education, medical care, social security, and other factors [29].

Simultaneously, according to communication theory, the spread of the Internet comprehensively affects society, not only influencing politics and economics, but also our way of life and mindset [30]. China’s social environment is rapidly disseminated through traditional media, such as TV, radio, newspapers, or new media, such as the Internet. In comparison, in terms of influence, the Internet quickly adapts to the need for acquiring information and communicating in the fast-paced life of the current society, thereby gradually replacing traditional media, such as the TV and newspapers, and becoming the main source of information release and dissemination [31].

Furthermore, according to information economics, to achieve optimal decision-making when rural residents participate in the social environment, they will search for information through various channels to eliminate information asymmetry [32]. Due to social learning, both the original news content and the comments of Internet users on the news will significantly affect users’ decision-making [23]. Owing to the limited screening ability of rural residents and the high cost of screening, a large volume of news about the social environment, whether authentic or not, may be acquired and absorbed by rural residents and then affect their behavior.

According to behavioral economics and grounded on the research of scholars, information dissemination on the Internet has both positive and negative effects. In other words, the Internet can positively influence people’s opinions on issues through publicity and education, creating a “media and publicity mobilization effect” [33,34,35]. Conversely, there may also be ineffective or counterproductive publicity, creating a “media and publicity boomerang effect” [35,36]. While rural residents are affected by many factors, such as their own backgrounds and information disseminated on the Internet, ultimately, they will have personal psychological expectations and make decisions about their satisfaction with the social environment that benefit them. Based on this, to study the mechanism of impact of Internet use on satisfaction with the social environment, this study postulates the following hypothesis:

**Hypothesis 1.** 
*Due to the presence of the “media and publicity mobilization effect” and “media and publicity boomerang effect,” Internet use to obtain news significantly affects rural residents’ satisfaction with the social environment.*


Based on the “Statistical Report on China’s Internet Development”, although China’s Internet penetration rate had reached 71.6% as of June 2021, the Internet penetration rate in rural areas was less than 60% and varied greatly. Many scholars have researched the fairness of information supply to address this issue, but the debate has not reached a consensus [37,38]. There remain obvious technicalities of the impact of historical institutional barriers and rapid social changes on the economic structure of different regions in rural China, and the resulting gap in regional development will inevitably lead to differences in the social environment of rural residents with different characteristics. This also means that there are bound to be differences in public Internet resources that people of different genders, educational backgrounds, ages, and regions can access, leading to differentiation [39]. The unfairness in accessing the supply of information resources mainly manifests in whether rural residents can freely acquire information as a resource with different characteristics. If rural residents with different characteristics can freely access information resources, it will reduce the problems of information asymmetry, waste, and rent-seeking, as well as the social cost of information acquisition for rural residents with different characteristics [40].

According to information economics, online communication represents the fastest-growing means of communication in human history, with evident “information dividends” [30]. It has increasingly become the main tool for public information search [5,22], with access opportunities and differences in use being the main influences of the digital divide [41]. The difference in use is a prominent factor that exacerbates the digital divide due to differences in the degree of Internet development in different regions [42,43]. Ruan believed that many rural residents had information facilities, but the awareness of information acquisition was not very strong and mainly derived from offline sources [44]. It seems that the unconditional optimism of the Internet has driven people towards Internet-centrism, obfuscating growing social injustice, knowledge monopolies, and even anti-democratic trends [45].

As McChesney noted, the new generation of information technology, such as the Internet, has not dismantled the monopoly of media, culture, and knowledge, nor has it significantly improved social equity [46]. In a modern society with the development of the Internet, improving the information supply is the only way to bridge the digital divide and enjoy “information dividends” [47]. Furthermore, researchers in relevant fields have found that the media’s influence on individuals is mediated by demographic factors [33,47]. For example, Geddes researched the influence of media on people with different education levels, while Shah et al. researched differences in media use across different ages or generations [48,49]. Therefore, considerable differences exist in influencing factors, such as individual characteristics and the information acquisition methods of different groups of rural residents, with heterogeneity issues, so comparative analyses should be carried out [50,51,52]. Hence, referring to the research of Zhang and Zhuang, this paper puts forward the following hypothesis [39,52]:

**Hypothesis 2.** 
*Due to the digital divide, the impact of Internet use to obtain news on satisfaction with the social environment of groups with different characteristics may differ.*


## 3. Data, Method, and Descriptive Statistics

### 3.1. Data Source

The data used in this article are derived from the China Family Panel Studies (CFPS), a large-scale, nationally representative micro-level household survey conducted by the Institute of Social Science Survey of Peking University. The CFPS2014 sample covered 25 provinces, with a total sample size of 37,418. As the research objective of this paper is the impact on rural residents’ satisfaction with the social environment across the country, after retaining key variables, such as social environment satisfaction, Internet use, individual characteristics of respondents, political leanings, objective environment, and regional characteristics, and further eliminating samples with missing values, 14,703 samples of rural residents were finally obtained.

### 3.2. Variable Selection

All variables were divided into three categories according to the research needs: explained variables, core explanatory variables, and control variables. The definitions and assigned values of the variables are shown in Table 1.

#### 3.2.1. Explained Variable

The explained variable in this study refers to the satisfaction with the social environment. The traditional method of using quantitative indicators, such as GDP, primary sector investment, and education investment, to estimate the economic and social development status is too one-sided [16,17]. With the development of the social economy, rural residents have the most intuitive sense of satisfaction with China’s social environment. The CFPS2014 research data use eight dimensions related to the vital interests of the public, such as government integrity, environmental protection, wealth inequality, employment, education, medical care, housing, and social security, as indicators of satisfaction with the social environment. The 8-dimensional questions in the CFPS are “In general, how seriously do you think government corruption/environmental issues... are in China?” The results were divided into 11 levels using the Likert scale, with 0 referring to not serious and 10 referring to very serious, and the respondent’s answer is selected as the actual value. (“Social security” refers to the social security system where the state and society distribute and redistribute national income through legislation and guarantees the basic living rights of members of society, especially those with special difficulties in life. In general, social security consists of social insurance, social relief, social welfare, and special care and placement.)

#### 3.2.2. Core Explanatory Variables

The core explanatory variable of this article is Internet use, and the question in the CFPS2014 about using the Internet to obtain news is “How many days did you learn about political information through Internet news in the past week?” The respondent’s answer is selected as the actual value. In addition, as traditional media will also affect residents’ value judgments, this paper also selected “how many days in the past week did you get political information from TV news” as the supplementary variable.

#### 3.2.3. Control Variables

Referring to previous studies, rural residents’ satisfaction with the social environment is easily affected by various factors, such as individual social characteristics, political orientation, and the objective environment. Therefore, control variables were set up from the following three aspects: (1) Due to the objective presence of individual differences, the individual characteristics of the interviewees will significantly affect their social environment satisfaction. The selected variables for gender, age, age-squared term, marital status, ethnicity, education level, health status, work status, social capital, and income status were used as indicators to measure the individual characteristics of respondents [15]; (2) Individual respondents have a high tendency to participate in politics, and a greater extent of active participation in political activities may have a significant impact on satisfaction with the social environment. This article selected participation in political groups and voting as indicators to measure an individual’s political participation tendency. Participation in political groups is reflected in the CFPS question, “Which of the following organizations are you currently a member of? Including the Chinese Communist Party, democratic parties, representatives of people’s congresses at and above the county/district level, members of Chinese People’s Political Consultative Conferences at and above the county/district level, trade unions, the Communist Youth League, All-China Women’s Federation, All-China Federation of Industry and Commerce, informal social organizations (communities, networks, salons, etc.), religious/belief groups, associations of private business owners, and associations of self-employed workers.” This question is a multiple-choice question, and the number of political groups the respondent participated in was selected as the value [28]; (3) Social capital may also affect rural residents’ satisfaction with the social environment; the variable “How good is the popularity relationship” in CFPS2014 was selected as the variable of social capital with 0 being the lowest and 10 the highest [53]; and (4) Due to the relative consistency between the asymmetry in information from the Internet and the economic development gradient among the eastern, central, and western regions, to exclude the regional differences of the interviewees, the regional variables were mainly used as dummy variables [54].

### 3.3. Descriptive Statistics

#### 3.3.1. General Descriptive Statistics of Rural Residents’ Use of the Internet and TV

Rural residents obtain information through various means. As shown in Table 2, 7826 residents obtained news through the TV, accounting for 53.22% of the sample size. As many as 3814 samples obtained political information from TV news seven days a week, accounting for 25.94% of the sample size, showing that the traditional media was still the main source of information for rural residents. In total, 2778 samples used the Internet to obtain news, while 1213 samples used the Internet seven days a week, accounting for 8.25%, which shows that the Internet is also an important channel for obtaining news. The number of people who never used the Internet to obtain news information was 34.34% more than those who did not use the TV, and the number of people who watched TV every day was 2601 more than those who used the Internet every day. In other words, more people learn news through the TV.

The use of the TV and the Internet was further analyzed, as shown in Figure 1. In total, 2135 samples used the TV and the Internet to obtain news at the same time, accounting for 14.52% of the sample size. The number of residents using only the TV and the Internet was 5691 and 643, respectively; 6234 neither used the TV nor the Internet and accounted for 42.40% of the total samples.

#### 3.3.2. Descriptive Statistics of Satisfaction with the Social Environment

To study the situation of the overall sample, 14,703 samples of rural residents were analyzed together with descriptive statistics (Table 3). Using the issue of government integrity as an example for analysis, 12,366 people scored above 5 points, accounting for 84.09% of the total number of people. For rural residents, the issue of government integrity in China was already serious, and government corruption may seriously affect people’s trust in the government [55]. Inefficient policies and institutions are ubiquitous, and inefficient policies and institutions were chosen because they served the interests of politicians or social groups in power at the expense of society as a whole [56,57].

Furthermore, as illustrated in the chart in Figure 2, China’s overall social and environmental problems were already rather serious, and the government needs to take effective measures to avoid mass incidents that endanger social stability and national security. Schultz, one of the key figures in institutional economics and a famous American economist, said, “Any system is a response to existing needs in real life” [58]. As China has entered the deep waters of reform, some existing political, economic, and other related systems cannot fully meet the needs of the contemporary era, and institutional innovation and improvement are needed to improve rural residents’ satisfaction with the social environment.

## 4. Impact of Internet Use on Social Environment Satisfaction

Based on the 14,703 samples of CPFS2014 survey data, an ordered probit model was used to analyze the overall impact of the Internet on the satisfaction with the social environment of rural residents, and the results are shown in Table 4.

As shown in Table 4, this paper focused on the impact of using the Internet to obtain news on rural residents’ satisfaction with the social environment. The regression results of the model show that the influence coefficient of Internet use is positive and significant at the 1% significance level, which means that as the number of days that rural residents used the Internet to obtain news per week increases, their satisfaction with China’s social environment worsens. This indicates that using the Internet to obtain news significantly negatively affects rural residents’ satisfaction with the social environment, which verifies hypothesis H1 of this study. Moreover, as the number of days that rural residents use the TV to obtain weekly news increases, their satisfaction with the social environment in terms of government integrity, environmental protection, wealth gap, and employment was also more negative but more positive regarding housing and social security. Therefore, the impact of Internet use has a larger coefficient than that of using the TV, affecting more dimensions of social and environmental satisfaction, and therefore verifying that with the development of the times, an increasing number of people are starting to use the Internet to obtain news.

For control variables, except in employment, gender differences significantly affected satisfaction with other dimensions of the social environment. A possible reason is that there were no significant gender differences between male and female rural residents in the sample of respondents engaged in agricultural production. Compared with women, men had lower satisfaction scores in terms of government integrity, environmental protection, and the inequality between rich and poor, while women had lower satisfaction scores in education, medical care, housing, and social security. This can be ascribed to men usually thinking more from a macro-level perspective, while women usually pay more attention to immediate interests in a patriarchal society.

The influence coefficient of the age variable was significant on the education dimension of social environment satisfaction of rural residents, indicating that there is a U-shaped relationship between age and educational satisfaction among rural residents. A plausible explanation may be that the older people themselves have a low level of education and live in the village all year round, so they do not have a strong perception of the role of education. On the contrary, younger people are more educated and knowledgeable, and they value education more.

Marital status significantly negatively affected the values for the poverty gap, employment, medical care, housing, and social security. Rural residents who had a spouse and started a family usually had stable and good material conditions. Therefore, they would have a better social environment than those without spouses.

Ethnic minorities had worse scores than Han Chinese in the six aspects, including environmental protection and inequality between the rich and the poor, which may be attributed to Han Chinese occupying China’s relatively good geography, locations, resources, policies, and other absolute advantages due to their relatively large proportion in the total population (about 92%).

The higher the education level of rural residents, the lower their satisfaction with China’s social environment, with intellectuals usually having stronger political demands. This is consistent with the research of Yang et al. [59]. In addition, healthy populations of rural residents had a lower inequality between the rich and the poor than unhealthy populations—this is in line with Maslow’s hierarchy of needs which states that when a person’s survival needs (health) are met, they inevitably pursue higher-level needs (social needs).

The influence coefficient of working status was negative, and it was evident that rural residents with stable jobs had a higher satisfaction with the social environment. Rural residents with higher incomes had fewer concerns about environmental protection and inequality between rich and poor but higher concerns about medical care. This is possibly because people with higher incomes have a greater awareness of environmental protection and inequality between the rich and the poor and, at the same time, have better medical services.

Rural residents with greater participation in organizations had worse scores on environmental protection issues, indicating that greater participation in organizational activities made it easier to understand the importance of environmental protection.

Generally, the correlation between rural residents’ social capital and their satisfaction with the social environment was significant. People with healthier relationships are more active in social activities, have stronger social networks, and have higher demands of the social environment.

Compared with rural residents in the eastern region, rural residents in the central region had more negative attitudes toward government integrity, environmental protection, wealth inequality, and employment. Due to a large number of land transfers for economic development, rural residents in the eastern region had lower social security than rural residents in the central region; compared with the eastern region, the western region was more deficient in its social security system. In addition, rural residents in the western region scored better than those in the eastern region in terms of government integrity, environmental protection, wealth gap, medical care, and social security, thereby reflecting that the social class was more stable in the western rural areas with a smaller gap between the rich and the poor.

## 5. Robustness Check

The principal component and factor analyses were carried out on the eight explained variables. Figure 3 is the scree pot diagram of the principal component analysis and factor analysis. As can be seen, the first and second characteristic roots were greater than 1, and the remaining characteristic roots were less than 1. Moreover, the KMO statistic of the sample was 0.8715. These demonstrated that the factor analysis method was more suitable.

According to the research of Lin and Du and the characteristic principle of the root being greater than 1, two common factors were extracted to represent each variable [60]. The classification of the principal component factors is shown in Table 5.

Then, the impact of Internet use on rural residents’ satisfaction with the social environment was analyzed separately, and the regression results are shown in Table 6. Models (1) and (3) are the regression results without adding control variables, while Models (2) and (4) are the regression results after adding control variables. The regression results show that the influence factors of the two core variables of the Internet and TV were significant, regardless of the addition of control variables.

The core of this paper focuses on the impact of Internet use to obtain news on rural residents’ satisfaction with the social environment. The regression results show that the influence coefficient of Internet use was positive and significant at the 1% significance level, showing that for rural residents who use the Internet to obtain news, as the number of days of Internet use per week increased, their satisfaction with both China’s social security status and social environment status worsened. For rural residents who use TV to obtain news, as the number of days of using the Internet per week increased, their satisfaction with the social environment status improved, but that with the social security status worsened. A possible reason is that, compared with the information disseminated by the Internet, the content reported on the social environment status in TV news has been screened and is more authentic and reliable.

From Table 6, it can be concluded that, compared with women, men had higher scores in terms of satisfaction with the social environment status but lower satisfaction with the social security status. This may be attributed to men usually considering economic issues from a more macro-level perspective, while women pay greater attention to their individual situations in a patriarchal society. There is a U-shaped relationship between age and social environment status among rural residents because most resources in society are in the hands of older people, while young people have just entered society and live a hard life, so young people pay more attention to the social environment status. Compared with those who had a spouse and started a family, people without a spouse experienced greater pressures in life and paid more attention to the macroeconomic situation, so they were less satisfied with the social environment status and are more pessimistic.

However, the satisfaction of the Han Chinese with the social environment status was higher than that of minority nationalities, which was mainly due to the absolute proportion of the Han nationality. The Han Chinese also enjoy absolute advantages, such as relatively good geography, location, resources, and policies. The higher the level of education, the lower the awareness of the social environment in China, with intellectuals usually having stronger political demands. Health status had no significant impact on either the social environment status or the social security status, while the influence coefficient of working status on the social security status was negative—people with stable jobs will have a better social security status. People with high incomes among rural residents were more satisfied with the social environment status and less satisfied with the social security status, reflecting that rural residents with higher incomes were less concerned about the macro-level situation but were dissatisfied with their social security status. Participating in organizations had no significant impact on the macro-social and social security status of rural residents. On the contrary, rural residents’ social capital had a significant impact on both.

Compared with rural residents in the eastern region, rural residents in the central region had a worse social security status, possibly resulting from the overall supporting facilities and systems in the rural areas of the central region not being able to meet the living needs of rural residents. Conversely, rural residents in the central region enjoyed a higher level of social environment status and were more optimistic about the overall social environment status than rural residents in the eastern region. Compared with rural residents in the eastern region, rural residents in the western region enjoyed a higher level of social security status, which could be because education, medical care, housing, and social security were relatively balanced in the western region, with contradictions not as significant as in the eastern region.

## 6. Discussion of Heterogeneous Effects

### 6.1. Heterogeneity Analysis of Rural Residents with Different Characteristics

As mentioned above, due to the imbalance in the social and economic development in China’s rural areas, rural residents with different characteristics had different perceptions of fairness, methods to access information, and the development of online informatization. When studying rural residents’ information acquisition methods and satisfaction with the social environment, if only the overall effect is considered, sample differences of groups with different characteristics will inevitably be ignored, and it will be difficult to have a reasonable understanding of the actual situation [54,61,62]. While online communication is the fastest-growing means of communication in the history of human beings, with an obvious “information dividend” [30] due to the unbalanced regional development, there are bound to be differences in public Internet resources that people of different genders, educational backgrounds, ages, and regions can obtain, causing a “digital divide” [39]. To explore this issue more comprehensively, this study analyzed and discussed the differences in the impact of Internet use to obtain news on the satisfaction with the social environment of rural residents with different characteristics.

To study sample differences between the rural residents of different genders, this section classified 14,703 rural residents, including 7242 female and 7461 male participants. Taking 45 years old as the cutoff to study age heterogeneity, there were 6619 samples of people younger than 45 years and 8084 samples of adults older than 45 years. To study the heterogeneity of academic qualifications, the data comprised 5010 samples with illiterate/semi-illiterate educational backgrounds and 9693 samples with a primary school education and above. Regional heterogeneity was studied by grouping different regions, including 5406 samples in the eastern region and 9297 samples in the central and western regions. From the descriptive statistics in Figure 4, it can be seen that women, middle-aged and older adults, and illiterate/semi-illiterate people used the Internet to obtain news less frequently.

### 6.2. Differences in the Effects of Using the Internet and TV on the Satisfaction with the Social Environment for Different Groups of People

As observed from Figure 5, in the regression results of population samples with different characteristics, there were no significant differences in the impact of Internet use to obtain news on the satisfaction with the social environment of rural residents of different genders, ages, and regions. Specifically, the Internet is a significant factor affecting the social environment satisfaction of male and female rural residents. If male and female rural residents use the Internet to obtain news information one day a week, their satisfaction with all social environments will be more negative; the coefficient of males is higher than that of females, which is significant at the 5% significance level. Social environment satisfaction is vulnerable to the significant impact of the Internet, whether it is for the youth, middle-aged, or elderly people. With the increase in time they spend on the Internet to obtain information every week, young, middle-aged, and elderly people have a more negative social environment satisfaction, and the impact coefficient of the Internet on the sample of middle-aged and elderly people is higher. The Internet is a significant factor affecting rural residents’ social environmental satisfaction in the eastern, central, and western regions. Moreover, the Internet has a greater impact on rural residents in the eastern regions. Internet use was a significant influencing factor for samples of rural residents with a primary school education and above, while it was not a significant influencing factor for illiterate/semi-literate rural residents, thereby verifying H2. In other words, this reflects a digital divide, which is consistent with Wang’s research [35].

As can be seen from Figure 6, in the sample regression results of different characteristic groups, the use of the TV to obtain news information has a significant impact on the social environment satisfaction of rural residents of different genders, ages, educational backgrounds, and regions. Consistent with the benchmark return, the number of days rural residents with different characteristics use the TV to acquire news each week has increased, and their satisfaction with the social environment in terms of government integrity, environmental protection, the gap between the rich and the poor, employment, and other aspects is more negative, but they are more positive in terms of education, medical care, housing, and social security. Specifically, when female rural residents increase the use of the TV to obtain news information one day a week it results in more negative social environment satisfaction in terms of government integrity, the gap between the rich and the poor, and employment. However, male rural residents who use the TV to obtain news information one day a week are more negatively satisfied with the social environment in terms of government integrity, environmental protection, and the gap between rich and poor, and more positively satisfied with the social environment in terms of housing and social security. The use of the TV is also a significant influencing factor for young people in terms of government integrity, environmental protection, the gap between rich and poor, and employment, while the use of the TV is a significant influencing factor for middle-aged and elderly people in terms of government integrity, environmental protection, the gap between rich and poor, education, housing, and social security. Rural residents with a primary school education and above use the Internet to obtain news information every day, and their satisfaction with the social environment in terms of government integrity, environmental protection, the gap between the rich and the poor, employment, etc. is more negative. Rural residents with illiterate/semi-illiterate educational backgrounds who use the TV to obtain news information every day are more negative in terms of social environment satisfaction regarding government integrity and the gap between rich and poor, and more positive regarding housing and social security. No matter which region, the use of the TV will affect the social environment satisfaction of rural residents in terms of government integrity, environmental protection, wealth gap, employment, education, housing, and social security.

## 7. Conclusions and Policy Implications

### 7.1. Conclusions

Based on the survey data from the CFPS2014, this study used an ordered probit model to conduct an empirical study on the impact of using the Internet to obtain news on rural residents’ satisfaction with the social environment. The conclusions are as follows:

Using the Internet to obtain news significantly affects rural residents’ social environment satisfaction and has a greater impact on social environment satisfaction than traditional media such as TV. This demonstrates that the Internet has gradually become the main source of information for rural residents. Individual characteristics, such as gender, age, ethnicity, and education level, also significantly affected satisfaction with the social environment. Men had lower satisfaction scores than women regarding government integrity, environmental protection, and the inequality between rich and poor, while women had lower satisfaction scores in education, medical care, housing, and social security. Ethnic minorities scored worse than Han Chinese in six aspects, including environmental protection and the inequality between rich and poor, while young people had lower social environment scores than older people. Furthermore, the higher the level of education among rural residents, the lower their satisfaction with China’s social environment. Women, middle-aged and older adults, and illiterate/semi-illiterate people used the Internet to obtain news less frequently. There were no significant differences in the impact of Internet use to obtain news on the social environment satisfaction of rural residents of different genders, ages, and regions. However, it significantly affected the satisfaction of the social environment of the population with an education level of primary school or above, but not the illiterate/semi-illiterate population, reflecting the emergence of a digital divide.

### 7.2. Policy Implications

In many developing countries, including China, the Internet has rapidly formed part of the basic infrastructure in vast rural areas. To better guide the spread of public opinions on the Internet and improve rural residents’ satisfaction with the social environment, the following suggestions are proposed based on the research conclusions of this study:

Attach great importance to the role of modern media, such as the Internet and TV, on rural residents’ satisfaction with the social environment. There should be increased efforts to deal with false public opinion, rumors, and negative publicity, purify the public information environment in rural areas, and include rural residents in advocating enhanced positive energy, thereby improving overall satisfaction with the social environment [63]. Increase the penetration rate of modern media such as the Internet and TV in rural areas, actively improve the Internet and digital literacy of vulnerable groups among rural residents, and improve and eliminate the digital divide. The government should speed up the development of informatization in rural areas, especially the construction of a basic network infrastructure, so that vulnerable groups in rural areas can obtain necessary information resources and rural residents can be encouraged to use the Internet for positive sources of information [64,65].

## Figures and Tables

**Figure 1 ijerph-20-01844-f001:**
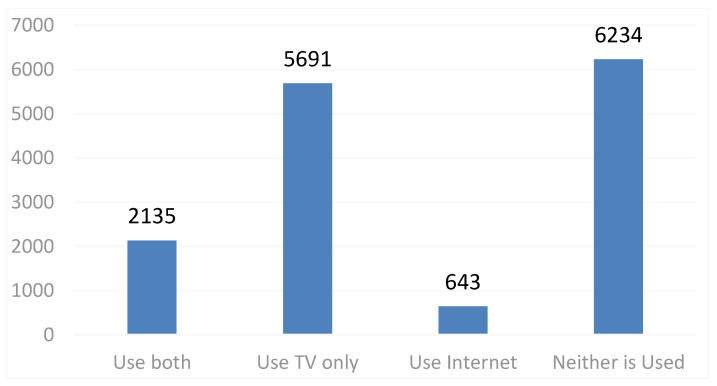
Rural residents’ use of TV and Internet.

**Figure 2 ijerph-20-01844-f002:**
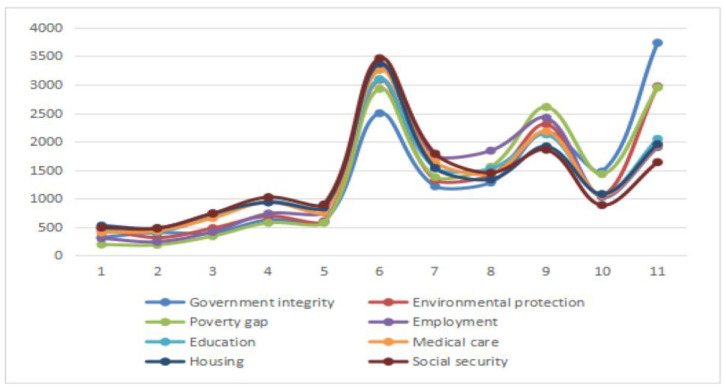
Diagram of the trends in rural residents’ satisfaction with the social environment.

**Figure 3 ijerph-20-01844-f003:**
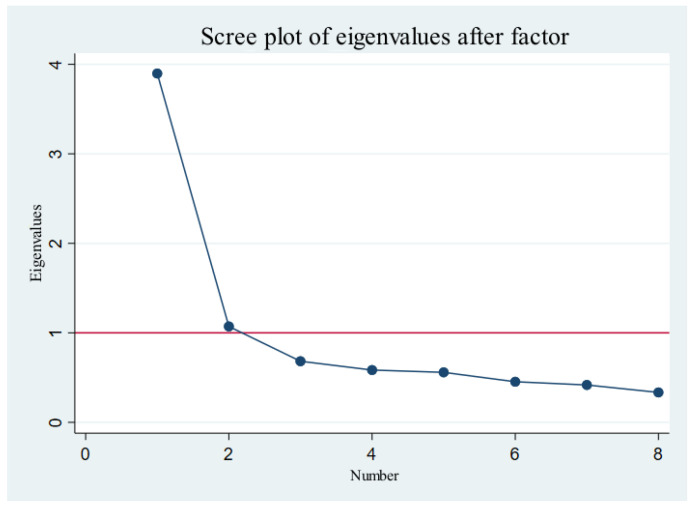
Scree plot diagram of the principal component analysis and factor analysis.

**Figure 4 ijerph-20-01844-f004:**
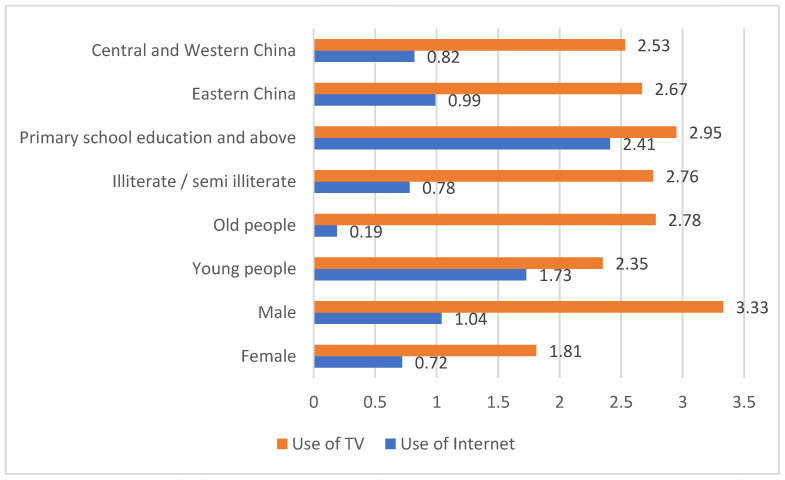
Comparison of the number of days that rural residents in different groups used the Internet and TV.

**Figure 5 ijerph-20-01844-f005:**
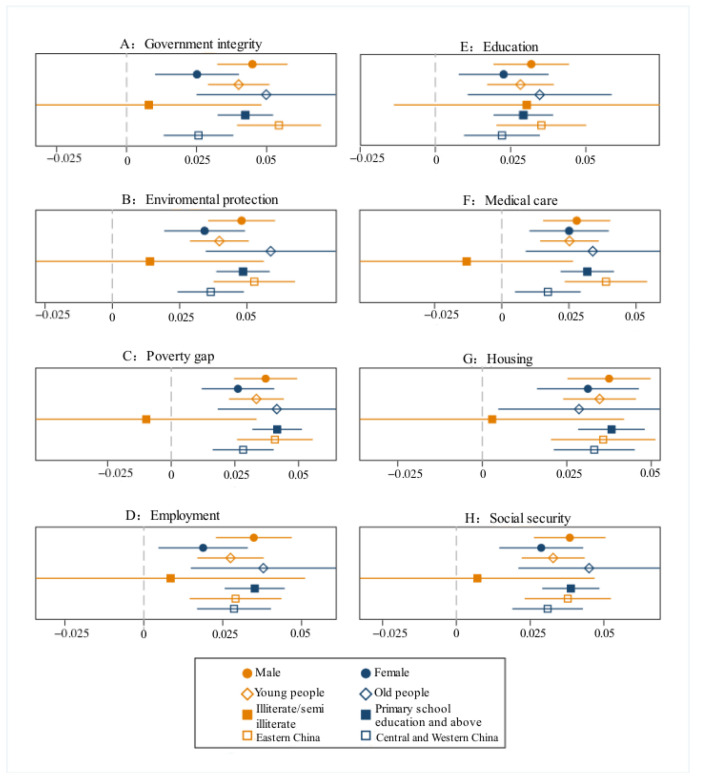
Differences in the impact of Internet use on the satisfaction with social environment among different groups.

**Figure 6 ijerph-20-01844-f006:**
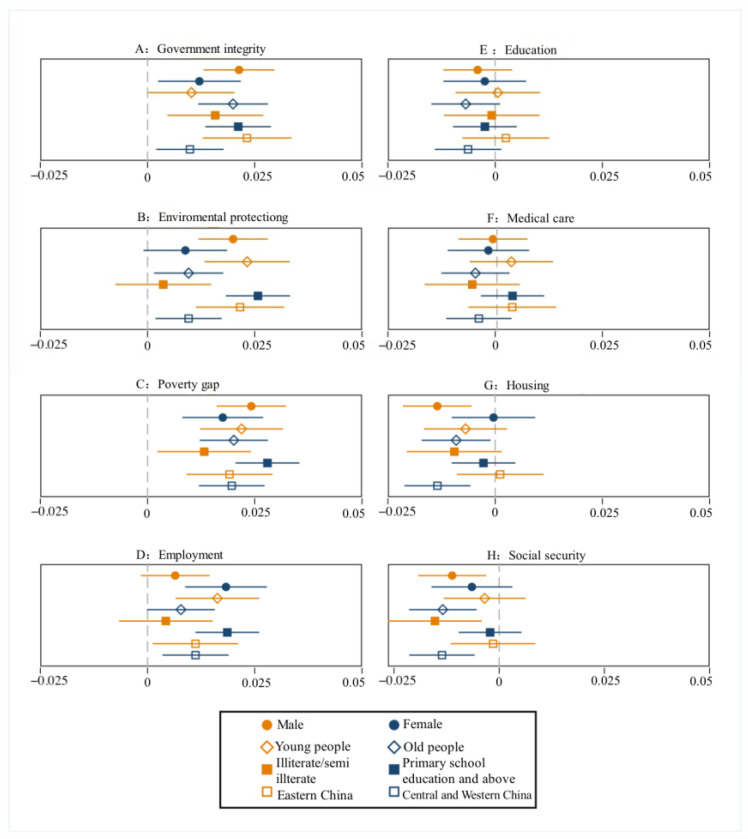
Differences in the impact of TV use on the satisfaction with social environment among different groups.

**Table 1 ijerph-20-01844-t001:** Value of variables.

Variables		Value	Mean	Standard Deviation	Min	Max
Explained variable	Government integrity	Actual score	6.9459	2.7118	0	10
	Environmental protection	Actual score	6.5702	2.6994	0	10
	Poverty gap	Actual score	6.9263	2.4291	0	10
	Employment	Actual score	6.3847	2.4113	0	10
	Education	Actual score	6.1464	2.6587	0	10
	Medical care	Actual score	6.1290	2.6216	0	10
	Housing	Actual score	5.9909	2.7074	0	10
	Social security	Actual score	5.8490	2.6112	0	10
Core explanatory variable	Use of Internet	Actual score	0.8827	2.0834	0	7
	Use of TV	Actual score	2.5847	2.9369	0	7
Control variables	Gender	Male = 1; Female = 0	0.5074	0.5000	0	1
	Age	Actual value	46.2989	15.7799	16	99
	Age square ^1^	The square of age	2392.5780	1513.6490	256	9801
	Marital status	Yes = 1; No = 0	0.8326	0.3733	0	1
	Ethnicity	Han = 1; Others = 0	0.8956	0.3058	0	1
	Education	Illiterate/semi-illiterate = 1; Primary school = 2; Junior middle school = 3; High school/technical secondary school/technical school/vocational school = 4; Junior college = 5; Undergraduate = 6; Master = 7; Doctor = 8	2.2281	1.1201	1	7
	Health	Very healthy = 1; Healthy = 2; Relatively healthy = 3; Generally healthy = 4; Unhealthy = 5	2.9595	1.2886	1	5
	Work	Yes = 1; No = 0	0.7918	0.4060	0	1
	Income	Logarithm ^2^	7.3782	2.2442	0	10
	Participating organizations	The total number of political groups	0.1724	0.3998	0	4
	Social capital	Actual score	7.2373	1.8560	0	10
	Central China	Dummy variable	0.2957	0.4564	0	1
	Western China	Dummy variable	0.3366	0.4726	0	1

^1^ The age square is added to observe whether there is an inverted U-shaped relationship between age and the explained variables. ^2^ The heteroscedasticity is not eliminated, and the logarithm of income is taken. However, because some samples have zero values, log (x + 1) is taken as the logarithm.

**Table 2 ijerph-20-01844-t002:** Statistical analysis of rural residents obtaining news through the Internet and TV.

	Days	0	1	2	3	4	5	6	7
Use of Internet	Number	11,925	276	426	411	161	263	28	1213
	Proportion	81.11%	1.88%	2.9%	2.8%	1.1%	1.79%	0.19%	8.25%
Use of TV	Number	6877	550	1228	1243	480	416	95	3814
	Proportion	46.77%	3.74%	8.35%	8.45%	3.26%	2.83%	0.65%	25.94%

**Table 3 ijerph-20-01844-t003:** Scores of rural residents’ satisfaction with the social environment.

	Level	0	1	2	3	4	5	6	7	8	9	10
Government integrity	Number	319	400	395	625	598	2497	1215	1281	2165	1473	3735
	Proportion	2.17%	2.72%	2.69%	4.25%	4.07%	16.98%	8.26%	8.71%	14.72%	10.02%	25.40%
Environmental protection	Number	474	314	479	691	597	3082	1326	1413	2302	1054	2971
	Proportion	3.22%	2.14%	3.26%	4.70%	4.06%	20.96%	9.02%	9.61%	15.66%	7.17%	20.21%
Poverty gap	Number	192	186	339	573	572	2926	1376	1556	2607	1428	2948
	Proportion	1.31%	1.27%	2.31%	3.90%	3.89%	19.90%	9.36%	10.58%	17.73%	9.71%	20.05%
Employment	Number	301	241	415	736	758	3314	1751	1840	2418	1027	1902
	Proportion	2.05%	1.64%	2.82%	5.01%	5.16%	22.54%	11.91%	12.51%	16.45%	6.98%	12.94%
Education	Number	436	412	677	955	833	3096	1532	1522	2125	1068	2047
	Proportion	2.97%	2.80%	4.60%	6.50%	5.67%	21.06%	10.42%	10.35%	14.45%	7.26%	13.92%
Medical care	Number	405	432	658	937	762	3258	1641	1438	2178	1059	1935
	Proportion	2.75%	2.94%	4.48%	6.37%	5.18%	22.16%	11.16%	9.78%	14.81%	7.20%	13.16%
Housing	Number	524	481	738	932	841	3,368	1536	1337	1915	1078	1953
	Proportion	3.56%	3.27%	5.02%	6.34%	5.72%	22.91%	10.45%	9.09%	13.02%	7.33%	13.28%
Social security	Number	488	476	738	1024	896	3464	1787	1452	1856	884	1638
	Proportion	3.32%	3.24%	5.02%	6.96%	6.09%	23.56%	12.15%	9.88%	12.62%	6.01%	11.14%

Note: Data are derived from the CFPS2014 survey data.

**Table 4 ijerph-20-01844-t004:** Benchmark regression results.

Variables	Government Integrity	Environmental Protection	Poverty Gap	Employment	Education	Medical Care	Housing	Social Security
Use of Internet	0.0392 ***	0.0432 ***	0.0351 ***	0.0292 ***	0.0261 ***	0.0259 ***	0.0340 ***	0.0338 ***
	(0.0050)	(0.0049)	(0.0049)	(0.0048)	(0.0048)	(0.0048)	(0.0048)	(0.0048)
Use of TV	0.0157 ***	0.0136 ***	0.0187 ***	0.0098 ***	−0.0037	−0.0017	−0.0091 ***	−0.0101 ***
	(0.0031)	(0.0031)	(0.0031)	(0.0031)	(0.0031)	(0.0031)	(0.0031)	(0.0031)
Gender	0.1045 ***	0.0336 *	0.0794 ***	−0.0028	−0.0672 ***	−0.0512 ***	−0.0355 *	−0.0330 *
	(0.0186)	(0.0185)	(0.0185)	(0.0183)	(0.0183)	(0.0183)	(0.0183)	(0.0183)
Age	0.0005	−0.0123 ***	0.0024	−0.0070 *	−0.0193 ***	−0.0100 ***	−0.0139 ***	−0.0119 ***
	(0.0038)	(0.0038)	(0.0038)	(0.0037)	(0.0037)	(0.0037)	(0.0037)	(0.0037)
Age square	−0.0001	0.0000	−0.0001 **	0.0000	0.0001 **	0.0000	0.0000	0.0000
	(0.0000)	(0.0000)	(0.0000)	(0.0000)	(0.0000)	(0.0000)	(0.0000)	(0.0000)
Marital status	0.0006	−0.0027	−0.0502 *	−0.0615 **	0.0341	−0.0587 **	−0.0865 ***	−0.0350 ***
	(0.0262)	(0.0260)	(0.0260)	(0.0257)	(0.0257)	(0.0257)	(0.0258)	(0.0257)
Ethnicity	0.0097	−0.0641 **	−0.0967 ***	−0.0409	−0.0605 **	−0.0714 **	−0.0301	−0.0804 ***
	(0.0291)	(0.0290)	(0.0290)	(0.0287)	(0.0287)	(0.0287)	(0.0287)	(0.0286)
Educational level	0.0455 ***	0.0944 ***	0.0782 ***	0.0860 ***	0.0341 ***	0.0436 ***	0.0323 ***	0.0345 ***
	(0.0093)	(0.0093)	(0.0093)	(0.0092)	(0.0091)	(0.0091)	(0.0092)	(0.0091)
Health	−0.0041	−0.0062	0.0193 ***	0.0033	−0.0021	0.0142 **	−0.0021	0.0061
	(0.0074)	(0.0073)	(0.0073)	(0.0072)	(0.0072)	(0.0072)	(0.0072)	(0.0072)
Work	0.0184	−0.0335	−0.0231	−0.0946 ***	−0.0270	−0.0006	−0.0212	−0.0373
	(0.0240)	(0.0239)	(0.0239)	(0.0237)	(0.0237)	(0.0237)	(0.0237)	(0.0236)
Income	0.0034	0.0057 **	0.0058 ***	−0.0019	−0.0027	−0.0042 *	0.0022	−0.0009
	(0.0022)	(0.0022)	(0.0022)	(0.0022)	(0.0022)	(0.0022)	(0.0022)	(0.0021)
Participating organizations	0.0164	0.0398 *	0.0090	−0.0017	−0.0130	−0.0343	0.0023	−0.0034
	(0.0215)	(0.0214)	(0.0214)	(0.0211)	(0.0211)	(0.0211)	(0.0211)	(0.0211)
Social capital	0.0197 ***	0.0338 ***	0.0383 ***	0.0402 ***	0.0292 ***	0.0312 ***	0.0367 ***	0.0293 ***
	(0.0047)	(0.0047)	(0.0047)	(0.0046)	(0.0046)	(0.0046)	(0.0046)	(0.0046)
Central China	0.1396 ***	−0.0700 ***	0.0696 ***	0.0525 **	0.0078	−0.0196	−0.0067	−0.0406 *
	(0.0215)	(0.0213)	(0.0213)	(0.0211)	(0.0211)	(0.0211)	(0.0211)	(0.0210)
Western China	−0.1109 ***	−0.1655 ***	−0.1334 ***	0.0144	0.0008	−0.0691 ***	0.0037	−0.0518 **
	(0.0211)	(0.0211)	(0.0210)	(0.0208)	(0.0208)	(0.0208)	(0.0208)	(0.0208)

Note: “*”, “**”, and “***” represent the 10%, 5%, and 1% significance levels, respectively, and the standard deviation is in brackets.

**Table 5 ijerph-20-01844-t005:** Classification of the principal component factors.

Variables	Principal Component Factor
Education; Medical care; Housing; and Social security	Social security status
Government integrity; Environmental protection; Poverty gap; and Employment	Social environment status

**Table 6 ijerph-20-01844-t006:** Regression results of the principal component and factor analyses.

Variables	Social Environment Status		Social Security Status	
	(1)	(2)	(3)	(4)
Use of Internet	0.0628 ***	0.0243 ***	0.0791 ***	0.0338 ***
	(0.0040)	(0.0045)	(0.0039)	(0.0045)
Use of TV	−0.0240 ***	−0.0145 ***	0.0317 ***	0.0248 ***
	(0.0028)	(0.0029)	(0.0028)	(0.0029)
Gender		−0.0901 ***		0.1047 ***
		(0.0175)		(0.0172)
Age		−0.0185 ***		0.0002
		(0.0036)		(0.0035)
Age square		0.0001 **		−0.0001 **
		(0.0000)		(0.0000)
Marital status		−0.0408 *		−0.0322
		(0.0245)		(0.0241)
Ethnicity		−0.0628 **		−0.0395
		(0.0274)		(0.0269)
Educational level		0.0239 ***		0.0988 ***
		(0.0087)		(0.0086)
Health		0.0043		0.0038
		(0.0069)		(0.0068)
Work		−0.0207		−0.0380 *
		(0.0225)		(0.0222)
Income		−0.0036 *		0.0061 ***
		(0.0021)		(0.0020)
Participating organizations		−0.0214		0.0234
		(0.0201)		(0.0198)
Social capital		0.0281 ***		0.0299 ***
		(0.0044)		(0.0043)
Central China		−0.0393 *		0.0866 ***
		(0.0201)		(0.0197)
Western China		0.0080		−0.1303 ***
		(0.0198)		(0.0195)
Constants	0.0066	0.5741 ***	−0.1517 ***	−0.3399 ***
	(0.0112)	(0.0864)	(0.0111)	(0.0850)

Note: “*”, “**”, and “***” represent the 0.1, 0.05, and 0.01 significance levels, respectively, and the standard deviations are in brackets.

## Data Availability

Research data can be obtained from the website: http://www.isss.pku.edu.cn/cfps/ (accessed on 27 December 2022).

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
