# Peer review of "Effect of Internet Use to Obtain News on Rural Residents’ Satisfaction with the Social Environment: Evidence from China"

_ijerph, 2023, doi:10.3390/ijerph20031844_

Round 1
Reviewer 1 Report
Using the CFPS2014 data, the present study assessed the impact of Internet use to obtain news on rural residents’ satisfaction with the social environment. This paper has important theoretical and practical significance, which may contribute to the literature in many ways. However, the paper in its current version suffers from some shortcomings. Below are my minor concerns about the paper.
1. The abstract content is too much, so it is recommended to do some cutting;
2. Social Network is an important channel for rural residents to obtain information. It proposes to select variables that can represent social capital as control variables;
3. Figures 5 and 6 would do well to add some descriptive interpretations;
4. Conclusions and policy implications can be short and more targeted.
Reviewer 2 Report
This paper studies the impact of Internet use on rural residents’ satisfaction with their social environment. Firstly, this paper checks the difference of rural residents’ use of TV and Internet. Then, it analyzes the scores of rural residents’ satisfaction with the social environment in 8 dimensions statistically. Next, the factors of rural residents’ satisfaction with the social environment are explored. Finally, it discusses the heterogeneous effects of Internet use on rural residents’ satisfaction among groups with different characteristics. In general, the research perspective has a certain novelty and the structure is basically reasonable. However, there still exist some drawbacks that need to be further addressed.
â‘ The study design, analysis process and results of this paper are relatively simple and easy.
â‘¡ The unreasonable setting of some indicators leads to inadequate and unreasonable explanations. For example, the explanation of the impact of age on rural residents’ satisfaction is unreasonable in lines 346-352. Authors stated in paper that “most resources in society are in the hands of older adults with higher levels of work experience and coupled with delayed retirement and other policies, wealth transfer is prolonged. However, young people have just entered society and live difficult lives, so they are inevitably pessimistic about the future.” The age range of the rural residents interviewed in this paper is from 16 to 102 years old, so I suggest to add the age square to the variables table, which may be helpful to explore the complex relationship between age and satisfaction. Besides, I do not think elderly people in rural China will be significantly affected by policies such as delayed retirement.
â‘¢ Figures 5 and 6 are not the most straightforward and concise way to express the regression results. Moreover, the lack of annotation of coordinate axis information on the graph leads to difficulty in reading the graph.
â‘£ The classification of principal component and factors in table 5 lacks the necessary clarification. Why the variables of Education, Medical care, Housing and Social security are classified to named as Micro-social environment? Why the variables of Government integrity, Environmental protection, Poverty gap and Employment are classified to named as Macro-social environment?
⑤ The suggestions put forward in part 7.2 should be more concrete.
â‘¥ More references published in recent 3 years should be cited.
